# Developing a Novel and Optimized Yeast Model for Human VDAC Research

**DOI:** 10.3390/ijms252313010

**Published:** 2024-12-03

**Authors:** Martyna Baranek-Grabińska, Wojciech Grabiński, Deborah Musso, Andonis Karachitos, Hanna Kmita

**Affiliations:** 1Department of Bioenergetics, Faculty of Biology, Institute of Molecular Biology and Biotechnology, Adam Mickiewicz University, 61-614 Poznań, Poland; martyna.baranek-grabinska@amu.edu.pl (M.B.-G.); wojciech.grabinski@amu.edu.pl (W.G.); 2Department of Molecular Medicine, University of Pavia, 27100 Pavia, Italy; deborah.musso01@universitadipavia.it

**Keywords:** human VDAC paralogs, *por1*Δ*por2*Δ mutants, yeast complementation assay, yeast strain genotype, *MET15*, cysteine-depleted variant of hVDAC3

## Abstract

The voltage-dependent anion-selective channel (VDAC) plays a crucial role in mitochondrial function, and VDAC paralogs are considered to ensure the differential integration of mitochondrial functions with cellular activities. Heterologous expression of VDAC paralogs in the yeast *Saccharomyces cerevisiae por1*Δ mutant cells is often employed in studies of functional differentiation of human VDAC paralogs (hVDAC1-hVDAC3) regardless of the presence of the yeast second VDAC paralog (yVDAC2) encoded by the *POR2* gene. Here, we applied *por1*Δ*por2*Δ double mutants and relevant *por1*Δ and *por2*Δ single mutants, derived from two *S. cerevisiae* strains (M3 and BY4741) differing distinctly in auxotrophic markers but commonly used for heterologous expression of hVDAC paralogs, to study the effect of the presence of yVDAC2 and cell genotypes including *MET15*, the latter resulting in a low level of hydrogen sulfide (H_2_S), on the complementation potential of heterologous expression of hVDAC paralogs. The results indicated that yVDAC2 might contribute to the complementation potential. Moreover, the possibility to reverse the growth phenotype through heterologous expression of hVDAC paralogs in the presence of the applied yeast cell genotype backgrounds was particularly diverse for hVDAC3 and depended on the presence of the protein cysteine residues and expression of *MET15*. Thus, the difference in the set of auxotrophic markers in yeast cells, including *MET15* contributing to the H_2_S level, may create a different background for the modification of cysteine residues in hVDAC3 and thus explain the different effects of the presence and deletion of cysteine residues in hVDAC3 in M3-Δ*por1*Δ*por2* and BY4741-Δ*por1*Δ*por2* cells. The different phenotypes displayed by BY4741-Δ*por1*Δ*por2* and M3-Δ*por1*Δ*por2* cells following heterologous expression of a particular hVDAC paralog make them valuable models for the study of human VDAC proteins, especially hVDAC3, as a representative of VDAC protein sensitive to the reduction–oxidation state.

## 1. Introduction

The voltage-dependent anion selective channel (VDAC) forms an omnipresent pathway for metabolite transport across the outer mitochondrial membrane [1,2,3,4,5,6,7]. The molecular mass cutoff for the pathway is approximately 4 kDa, and the transported molecules range from inorganic ions (e.g., K^+^, Na^+^ and Cl^−^) to metabolites of different sizes and charges (e.g., large anions, such as ATP, AMP, NADH and glutamate; small anions, such as superoxide anion; and large cations, such as acetylcholine) and large macromolecules, such as tRNAs [7,8,9]. The magnitude of transport through VDAC can be limited when VDAC switches to lower conducting substates featuring less anion selectivity [3], and the process is influenced by the presence of VDAC paralogs [10,11], as well as their post-translational modifications and interactions with different proteins [1,12].

The yeast *Saccharomyces cerevisiae* complementation assay is a useful method for assessing the ability of heterologously expressed proteins to reverse the growth phenotype of the relevant mutant strain, and has also been used for VDAC proteins from different organisms [13,14,15,16]. Yeast mitochondria contain two VDAC paralogs, yVDAC1 and yVDAC2, encoded by the *POR1* and *POR2* genes, respectively [17,18]. These paralogs differ dramatically in their expression levels. Namely, yVDAC1 has been shown to be expressed at levels five orders of magnitude greater than yVDAC2, and the difference was observed in the presence of a fermentable or nonfermentable carbon source (glucose and glycerol, respectively, the latter being metabolized by respiration that requires functioning mitochondria) [19]. Accordingly, it has been estimated that yVDAC1 is responsible for approximately 90% of the permeability of a single mitochondrial outer membrane [20]. In addition, yVDAC2 cannot compensate for the lack of yVDAC1 unless it is expressed under the control of the yVDAC1 promoter as far as growth of relevant yeast cells is concerned [17]. Therefore, mutants depleted of only yVDAC1 are used in studies on VDAC paralogs based on heterologous expression in *S. cerevisiae* cells.

The single mutants are derived from commonly used *S. cerevisiae* strains (for the relevant genotypes, see Table 1) including M3 [12,14,18,21,22,23,24], as well as closely related BY4741 [19,21,25,26] and BY4742 [13,19,27,28]. The M3 strain is the isogenic strain for the *por1*Δ mutant termed M22-2 and was obtained by Blachly-Dyson et al. (1997) [17]. It was generated by deleting most of the *POR1* gene through the insertion of the *LEU2* auxotrophic marker. BY4741-Δ*por1* and BY4742-Δ*por1* mutants are commercially available (Euroscarf, Frankfurt, Germany) and were obtained through the application of a G418 resistance cassette (KanMX4) for the deletion of the *POR1* gene in BY4741 or BY4742 isogenic wild-type cells, respectively. To verify the functionality of human VDAC paralogs, M3-Δ*por1* and BY4742-Δ*por1* mutant strains were used [13,14,27], but not BY4741- Δ*por1*.

As reported in other mammals, three human VDAC paralogs (hVDAC1, hVDAC2, and hVDAC3) have been identified [4,11,29]. It is generally assumed that hVDAC1, hVDAC2, and hVDAC3 are structurally very similar, but subtle sequence changes may facilitate paralog-specific roles through unknown mechanisms [27]. Accordingly, hVDAC1 and hVDAC2 are expressed at higher levels than hVDAC3, which is far less abundant, with the exception of levels observed in the testis, kidney, brain, heart, and skeletal muscle [30]. Nevertheless, heterologous expression of these paralogs in the mentioned M3-Δ*por1* and BY4742-Δ*por1* mutant cells using relevant pYX212 plasmid constructs [13,14,27] has indicated that hVDAC1, hVDAC2, and hVDAC3 form stable, highly conductive voltage-gated channels that are weakly anion-selective and facilitate metabolite exchange [14,27]. However, it has also been shown that the gating of hVDAC3 requires a reduction in disulfide bonds formed by specifically localized cysteine residues and the linking of the N-terminal region of the protein to the bottom of the channel pore [31]. Thus, hVDAC3 channel activity in the mitochondria is likely controlled by the protein functioning as redox sensor [4,31,32,33]. This particular property of hVDAC3 may explain why heterologous expression of hVDAC1 and hVDAC2, but not of hVDAC3, complements the growth defect of M3-Δ*por1* and BY4742-Δ*por1* mutant cells under restrictive conditions (37 °C in the presence of glycerol) [13,14,34] known to result in oxidative stress, e.g., [35,36].

The diverse content of cysteine residues is regarded as a distinctive feature of human VDAC paralogs, i.e., hVDAC1 has two, hVDAC2 has nine, and hVDAC3 has six. The residues follow an evolutionarily conserved oxidative modification pattern, being oxidated or reduced probably depending on their location with respect to cytosol or the intermembrane space [32,37,38]. However, only in hVDAC3 is the whole set of cysteine residues never detected as totally oxidized [39]. This suggests that hVDAC3 cysteine residues may undergo continuous reduction–oxidation (redox) cycles that in turn strengthen the assumption that the residues are indispensable due to the protein’s ability to counteract oxidative stress, e.g., [38,39]. This may correlate with changes in the protein channel activity [38,40], which may possibly be influenced by interactions with cytosolic proteins [27].

Accordingly, it has been shown that *S. cerevisiae* is a convenient model to investigate the functional relationship between VDAC, redox states of cell compartments, and the expression levels and/or activity of cellular proteins [41]. It has also been shown that the intracellular redox states are distinctly influenced by both yeast VDAC paralogs [41], forming channels of comparable electrophysiological characteristics [22], but providing different permeability across the mitochondrial outer membrane [18]. It is also known that hydrogen sulfide (H_2_S) is an important gaseous signaling molecule that is critically involved in regulating redox homeostasis in eukaryotic cells [42]. The proposed mechanisms include, among others, the regulation of antioxidative pathways and the attenuation of cellular ROS levels [43], but H_2_S has never been considered in the context of VDAC regulation. Importantly, the *S. cerevisiae* BY4741 strain differs from the related BY4742 strain in only one auxotrophic marker, namely, *MET15,* which is absent in BY4741 but present in BY4742 (Table 1). The gene *MET15* (also known as *MET17*) is also present in M3 cells, and it encodes the enzyme O-acetyl homoserine-O-acetyl serine sulfhydrylase, which is responsible for incorporating sulfide along with O-acetylhomoserine into homocysteine. Loss of this activity in *S. cerevisiae* cells results in the production of high H_2_S levels [44,45].

Thus, to check the putative relationship between the absence of H_2_S and the inability of hVDAC3 to complement the growth defect of M3-Δ*por1* and BY4742-Δ*por1* mutant cells under restrictive conditions [13,14], we performed heterologous expression of hVDAC1-hVDAC3 in BY4741 cells in the presence or absence of *MET15*. Moreover, double *por1*Δ*por2*Δ mutants were used in the studies to eliminate the possible effect of yVDAC2. The results demonstrate that the background of the yeast cell genotype can be decisive for complementation of the absence of yeast VDAC paralogs by human VDAC paralogs, which was clearly observed for hVDAC3. Moreover, the results indicate that cysteine residues contribute to hVDAC3-mediated complementation of *por1*Δ*por2*Δ mutant growth defects, and this contribution may be partially related to *MET15* activity and the resulting H_2_S level.

## 2. Results

### 2.1. The Phenotypes of the por1Δpor2Δ Double Mutants Derived from the M3 and BY4741 Strains Are Similar Under Restrictive Conditions

To determine whether M3- and BY4741-derived mutants depleted of both yVDAC paralogs (M3-*por1*Δ*por2*Δ and BY4741-*por1*Δ*por2*Δ, see Table 1) display the same phenotype under known restrictive conditions (37 °C and the presence of glycerol), the BY4741-derived double mutant was constructed using CRISPR/Cas9 (Figure 1) to eliminate *POR1* (see Section 4).

The growth patterns of both double mutants, along with their respective single mutant equivalents and corresponding wild-type cells, were evaluated at permissive (28 °C) and restrictive (37 °C) temperatures in media containing either glucose (YPD) or glycerol (YPG). This evaluation was performed using the cell viability assay after serial dilution and plating of the samples (Figure 2A). Quantitative analysis of the assay (Figure 2B–E) indicated that neither the *por1*Δ mutants nor the *por1*Δ*por2*Δ mutants grew under the known restrictive conditions, which is in agreement with available data obtained for M3-*por1*Δ and M3-*por1*Δ*por2*Δ mutants [17] and is confirmed in the study. Moreover, under the specified conditions, *por2*Δ mutant strains exhibited growth patterns similar to those of the relevant isogenic wild-type strain. The similarity in growth was also observed for the *por1*Δ mutants across both strains. However, the BY4741-*por1*Δ*por2*Δ mutant displayed distinct growth on glycerol at permissive temperatures and weak growth on glucose at restrictive temperatures, which was not observed for the M3-*por1*Δ*por2*Δ mutant.

### 2.2. The por1Δpor2Δ Double Mutants Derived from the M3 and BY4741 Strains Differ in Complementation upon Heterologous Expression of Human VDAC Paralogs

To investigate whether an identical phenotype under restrictive conditions would result in similar effects of heterologous expression of human VDAC paralogs in both *por1*Δ*por2*Δ mutants, we evaluated the capacity of the different hVDAC paralogs to complement the growth defect observed under the applied conditions (Figure 3A). As indicated by quantitative analysis of the performed assay (Figure 3B–E), the expression of the paralogs in BY4741-*por1*Δ*por2*Δ mutant cells enhanced growth to a very similar extent, and the conditions applied did not differentiate between the paralogs. Conversely, in the M3-*por1*Δ*por2*Δ mutant, heterologous expression of hVDAC3 resulted in the weakest complementation, including a complete lack of yeast cell growth at restrictive temperatures, regardless of the presence of glycerol or glucose. Moreover, hVDAC2 expression most significantly enhanced growth in the M3-*por1*Δ*por2*Δ mutant.

Based on the assumption that hVDAC3 channel activity is dependent on its cysteine residues [27,31,40], we also examined the effect of heterologous expression of a cysteine-depleted variant of hVDAC3 (hVDAC3ΔCys). As shown in Figure 3, hVDAC3ΔCys reduced the growth of the BY4741-*por1*Δ*por2*Δ mutant but increased the growth of the M3-*por1*Δ*por2*Δ mutant under the specific conditions of the cell viability assay. This reduction was most pronounced at restrictive temperatures regardless of the carbon source, whereas the increase was most pronounced at permissive temperatures regardless of the carbon source.

### 2.3. The Effect of Cysteine Depletion in hVDAC3 on Yeast Cell Growth May Be Related to the Activity of Met15

Inactivation of *MET15* (encoding O-acetyl-homoserine-O-acetyl-serine sulfhydrylase) in *S. cerevisiae* leads to increased H_2_S production [44], as schematically shown in Figure 4A. As mentioned in the Introduction, H_2_S plays a critical role in maintaining the intracellular redox balance and promotes cell growth [46]. Therefore, we aimed to restore *MET15* presence in a BY4741-*por1*Δ*por2*Δ mutant expressing hVDAC3 or hVDAC3ΔCys and to evaluate the effect of *MET15* presence on cell growth. This was achieved by transforming BY4741-*por1*Δ*por2*Δ-hVDAC3 and BY4741-*por1*Δ*por2*Δ-hVDAC3-ΔCys mutant cells with the pUM plasmid, which also provided the *URA3* marker for selection of subsequent transformants (Figure 4B). The presence of *MET15* decreased the ability of hVDAC3 to reverse the BY4741-*por1*Δ*por2*Δ growth phenotype, but also attenuated the effect of hVDAC3ΔCys on the mutant cells when compared to the effect of hVDAC3 (Figure 4C,D). Moreover, this attenuation was particularly pronounced at restrictive temperatures for both glucose and glycerol.

## 3. Discussion

Here, we report for the first time the effects of heterologous expression of human VDAC paralogs in *S. cerevisiae* double *por1*Δ*por2*Δ mutants derived from two different yeast strains. The results show that identical phenotypes of mutants derived from different yeast strains under the specified conditions do not exclude significant differences in the effects of heterologous expression of a given protein. Furthermore, our results add to the ongoing discussion about the role of cysteine residues in hVDAC3 functionality and the possible mechanism of protein modulation.

We focused on two different and commonly applied yeast strains, M3 [12,14,18,21,22,23,24] and BY4741, as well as relevant *por1*Δ*por2*Δ mutants. Both mutants were unable to grow under diagnostic restrictive conditions (i.e., 37 °C, in the presence of glycerol), as reported previously for M3-*por1*Δ*por2*Δ [17]. However, despite the similar growth phenotypes under diagnostic restrictive conditions, M3-*por1*Δ*por2*Δ and BY4741-*por1*Δ*por2*Δ differed in terms of growth at restrictive temperatures in the presence of glucose and at permissive temperatures (28 °C) in the presence of glycerol. The observed differences, namely, the growth of the BY4741-*por1*Δ*por2*Δ mutant and the lack of the growth of the M3-*por1*Δ*por2*Δ mutant, are likely due to the initial genotypes of the strains.

Moreover, the observed cell growth differences between the double mutant BY4741-*por1*Δ*por2*Δ and the single mutant BY4741-*por1Δ* reveal interesting genetic interactions that shed light on the functional relationship between *POR1* and *POR2*. According to Collins et al. (2010) [47], genetic interactions can be positive (suppressive) or negative (synthetic sick/lethal), depending on how mutations affect cellular processes. The positive genetic interaction observed for cells of the BY4741 strain suggests that the absence of *POR2* attenuates the deleterious effects caused by the deletion of *POR1*. This could be due to a compensatory mechanism whereby the complete loss of VDAC channels causes the cell to activate alternative metabolic pathways or stress responses to mitigate the impact on growth. In other words, the cell may reprogram its metabolism to compensate for the loss of VDACs. Conversely, in cells of the M3 strain, the negative genetic interaction occurs because the relevant double mutation is more deleterious than the single *POR1* mutation that indicates that *POR1* and *POR2* have independent but complementary roles that are critical for the cell growth of the strain. The lack of effective compensatory mechanisms in cells of the M3 strain could exacerbate the negative effects of the loss of both VDAC channels. Nevertheless, the lack of growth of both *por1*Δ*por2*Δ mutants under restrictive diagnostic conditions allowed for the use of these cells for heterologous expression of human VDAC paralogs.

In the case of the BY4741-*por1*Δ*por2*Δ mutant, heterologous expression of hVDAC1, hVDAC2, and hVDAC3 uniformly enhanced cell growth regardless of the applied environmental conditions. Conversely, for the M3-*por1*Δ*por2*Δ mutant, heterologous expression of hVDAC3 exhibited the least effective complementation, including a complete absence of yeast cell growth at restrictive temperatures, independent of the presence of glycerol or glucose. Because the double mutants were generated from M3 and BY4741 strains lacking the *POR2* gene, the differential contribution of yVDAC2 can be excluded. The importance of the presence of yVDAC2 for the effect of human VDAC paralog heterologous expression may be illustrated by the observation that hVDAC2 expression most significantly enhanced the growth of the M3-*por1*Δ*por2*Δ mutant (this study), but expression of hVDAC1 is the most effective in the case of M3-*por1*Δ [14]. Nevertheless, two hypotheses could explain the lack of complementation of the M3-*por1*Δ*por2*Δ growth phenotype under the restrictive temperature by hVDAC3: (1) thermal instability of the isolated protein, evidenced by a melting temperature of 29 °C [27], which occurs in M3 cells but not in BY4741 cells; and/or (2) the oxidation state of cysteine residues, critical for hVDAC3 gating in reconstitution experiments [31] (Figure 5) or other mechanisms underlying the possibility of complementation of *por1*Δ mutant growth [40] that may differ between M3 and BY4741 cells. In addition, the cysteine residues of hVDAC3 have been identified as essential for the ability of the protein to counteract oxidative stress in human HAP1 cells [38]. It was also shown that hVDAC3ΔCys forms typical VDAC channels without affecting the rate of proper channel insertion, although its interactions with cytosolic proteins are altered [27].

Given that the replacement of hVDAC3 with hVDAC3ΔCys, a variant of hVDAC3 in which all cysteine residues are replaced by alanine, affected the growth of M3-*por1*Δ*por2*Δ and BY4741-*por1*Δ*por2*Δ cells differently, it can be concluded that cysteine residues play an important role in the complementation effect of hVDAC3. However, this effect is dependent on intracellular conditions related to the yeast cell genotype. In particular, one of the auxotrophic markers that distinguishes BY4741 from M3 strains is *MET15*, which is absent in BY4741 cells. The absence of *MET15* is known to lead to increased H_2_S production [44,45], which is considered important for mitigating oxidative stress [43,48]. Therefore, the difference in the set of auxotrophic markers may create a different background for hVDAC3 cysteine modification [37] and, thus, explain the different effects of hVDAC3 cysteine residue depletion in M3*-por1*Δ*por2*Δ and BY4741-*por1*Δ*por2*Δ cells. Essentially, if the absence of *MET15* is associated with the complementation effect of hVDAC3 and hVDAC3ΔCys in BY4741-*por1*Δ*por2*Δ cells, then *MET15* expression should influence the ability of the proteins to affect the growth phenotype of BY4741-*por1*Δ*por2*Δ cells. Indeed, *MET15* expression decreased the ability of hVDAC3 to reverse the growth phenotype, but increased the ability in the case of hVDAC3ΔCys expression, which makes the BY4741-*por1*Δ*por2*Δ mutant similar to the M3-*por1*Δ*por2*Δ mutant, although the relevant changes in the growth of BY4741-*por1*Δ*por2*Δ cells are statistically significant only at restrictive temperatures and more pronounced in the presence of glucose. The reason could be a temperature-related increase in ROS level [36], mutual interactions between the ROS level and glycolysis rate [49], and the effectiveness of hVDAC3 in counteracting the increase in ROS, depending on the presence of cysteines and *MET15* [4,38].

## 4. Materials and Methods

### 4.1. Plasmids

The pML104-POR1 plasmid, which was used for CRISPR/Cas9 genome editing, was derived from the pML104 plasmid, which was kindly provided by John Wyrick. This plasmid is cataloged with Addgene under the identifier #676380 [50]. For the generation of pML104-POR1, a specific guide RNA sequence (5′-GTTGTTCAATGTAGCGCCCA-3′) was integrated into the pML104 plasmid using the Q5^®^ Site-Directed Mutagenesis Kit from New England Biolabs following the protocol outlined by Hu et al. (2018) [51]. Genes encoding hVDAC 1, 2, and 3 and a cysteine-depleted variant of hVDAC3 (hVDAC3ΔCys) optimized for yeast codon usage were synthesized along with sequences flanking the *POR1* gene both upstream and downstream and cloned into pBSK(+) Simple-Amp or pBluescript II SK(+) plasmids from Biomatik (Kitchener, ON, Canada). The pUM plasmid was a gift from Markus Ralser. This plasmid has been registered with Addgene under the identifier #64176 [52]. This plasmid was designed to restore prototrophy in *Saccharomyces cerevisiae* strains lacking the *URA3* and *MET15* genes. The list of plasmids used in this research is shown in Table 2.

### 4.2. Strains and Culture Media

The NEB^®^ 5-alpha competent *E. coli* bacterial strain, purchased from New England Biolabs (Catalog # C2987H), was used for plasmid amplification. The bacterial culture was performed in liquid LB medium composed of 1% tryptone, 0.5% yeast extract, and 1% sodium chloride, with ampicillin added for a final concentration of 100 μg/mL. The culture was maintained in the dark at 37 °C.

The yeast strains used in this study are listed in Table 1. Several media were used to grow the yeast cells to meet the specific experimental requirements. YPD (1% yeast extract, 2% peptone, and 2% D-glucose) and YPG (1% yeast extract, 2% peptone, and 3% glycerol, pH adjusted to 5.5) media were used for the viability assay. In addition, SD-Ura (0.67% yeast nitrogen base without amino acids, yeast synthetic nutrient supplement without uracil, and 2% D-glucose) and SG-Ura (0.67% yeast nitrogen base without amino acids, yeast synthetic nutrient supplement without uracil, and 3% glycerol, pH adjusted to 5.5) media were also used for the viability assay, specifically for cells harboring the pUM plasmid. SD-Ura medium served a dual purpose. In addition to the viability assay, it was also used for the selection of cells containing the pUM and pML104-*POR1* plasmids. SDC+5-FOA (0.67% yeast nitrogen base without amino acids, complete yeast synthetic drop-out medium supplemented with 2% D-glucose, and 0.1% 5-fluoroorotic acid) medium supplemented with 5-fluoroorotic acid (5-FOA) was used to selectively remove plasmids from yeast cells. Agar at a concentration of 2% was added to the solid media.

### 4.3. Yeast Genetic Modification

The BY4741-*por1*Δ and BY4741-*por1*Δ*por2*Δ strains were generated using the previously described CRISPR/Cas9 method [50] with the pML104-POR1 vector together with repair DNA synthesized by annealing two complementary oligonucleotides, POR1_KO_1 and POR1_KO_2. These oligonucleotides were specifically designed to contain sequences flanking the *POR1* open reading frame by 55 nucleotides. Similarly, all strains expressing human VDACs were generated using CRISPR/Cas9 and homology-directed repair, with the pML104-POR1 vector and repair DNA produced by PCR amplification from plasmid templates (Figure 1). The sequences of all the repair DNA used are available in the Appendix A.

Yeast cells were transformed with 250 ng of pML104-POR1 vector and 400 ng of repair DNA using the Yeastmaker™ Yeast Transformation System 2 (Takara). After transformation, the cells were plated on SD-Ura solid medium and incubated in the dark at 28 °C for 3–5 days. The genetic modification of each strain was confirmed with PCR genotyping using targeted primers [53], followed by analysis of the PCR products using Sanger sequencing. Once the desired genome editing was confirmed, the pML104-POR1 plasmid was removed by culturing the cells on SDC medium supplemented with 5-FOA. All oligonucleotides are listed in Table 3.

### 4.4. Analysis of Cell Growth

Yeast growth was analyzed using a yeast viability assay. Yeast cells were grown in liquid YPD medium until they reached an optical density (OD) of 0.5. Serial dilutions of 10^−1^, 10^−2^, and 10^−3^ were then prepared. Then, 10 µL from each dilution was added to plates containing one of the following solid media: YPD, YPG, SD-Ura, or SG-Ura. The plates were incubated in the dark at 28 °C and 37 °C for 3 days. After incubation, the plates were scanned to obtain high-resolution images of the colonies. The images were then analyzed using ImageJ software (version IJ 1.46r) to quantify colony growth by converting the images to grayscale and then analyzing the pixel intensity values. To evaluate yeast growth, three independent experiments were performed.

### 4.5. Statistical Analysis

Statistical analysis was performed using GraphPad Prism version 10.2.1, and 2-way ANOVA was used to evaluate the data. Statistical significance is indicated by the following symbols: **** for *p* < 0.0001, *** for *p* < 0.001, ** for *p* < 0.01, * for *p* < 0.05, and ‘ns’ for not significant.

## 5. Conclusions

In conclusion, our study demonstrates that the cysteine residues of hVDAC3 are crucial for its ability to complement growth defects in yeast strains lacking endogenous VDAC proteins, with this effect being modulated by the yeast’s genetic background—particularly the presence or absence of the *MET15* gene. The differential growth responses observed between M3-*por1*Δ*por2*Δ and BY4741-*por1*Δ*por2*Δ cells upon expression of hVDAC3 and its cysteine-depleted variant (hVDAC3ΔCys) suggest that hydrogen sulfide (H₂S) production plays a significant role in modulating hVDAC3 function. Specifically, in *met15*Δ strains like BY4741, increased H₂S production may enhance the activity of hVDAC3 by preventing disulfide bond formation involving cysteine residues, thereby influencing the hVDAC3 gating.

Our findings highlight H₂S as an emerging molecule important for the modulation of hVDAC3 involvement in the cellular response to oxidative stress, acting through modifications of cysteine residues. This opens new avenues for future research to explore the molecular mechanisms underlying this modulation. Understanding these mechanisms could provide valuable insights into the regulation of VDAC3 in eukaryotic cells and its potential role in conditions associated with oxidative stress. Our proposed hypothesis, illustrated in Figure 5, suggests that the genetic background of yeast strains, particularly regarding the *MET15* gene, can significantly influence the phenotypic outcomes related to hVDAC3 expression by affecting VDAC3 gating mechanisms. Further studies are needed to validate this hypothesis and to investigate the potential applications of modulating VDAC3 activity in therapeutic contexts.

## Figures and Tables

**Figure 1 ijms-25-13010-f001:**
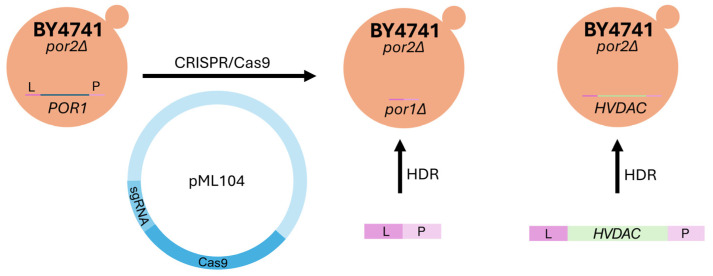
Creating model for human VDAC expression initially using *por2*Δ mutants (derived from the BY4741 strain). The vector pML104, containing Cas9 and sgRNA, is responsible for the CRISPR/Cas9 system. CRISPR/Cas9 creates a double-strand break (DSB) in the *POR1* locus. Subsequently, the repair DNA, through homologous recombination (HDR), uses homologous arms (L and P) to either generate a knockout (KO) mutant for the *POR1* gene (*por1*Δ) or to introduce the human *HVDAC* gene (knock-in).

**Figure 2 ijms-25-13010-f002:**
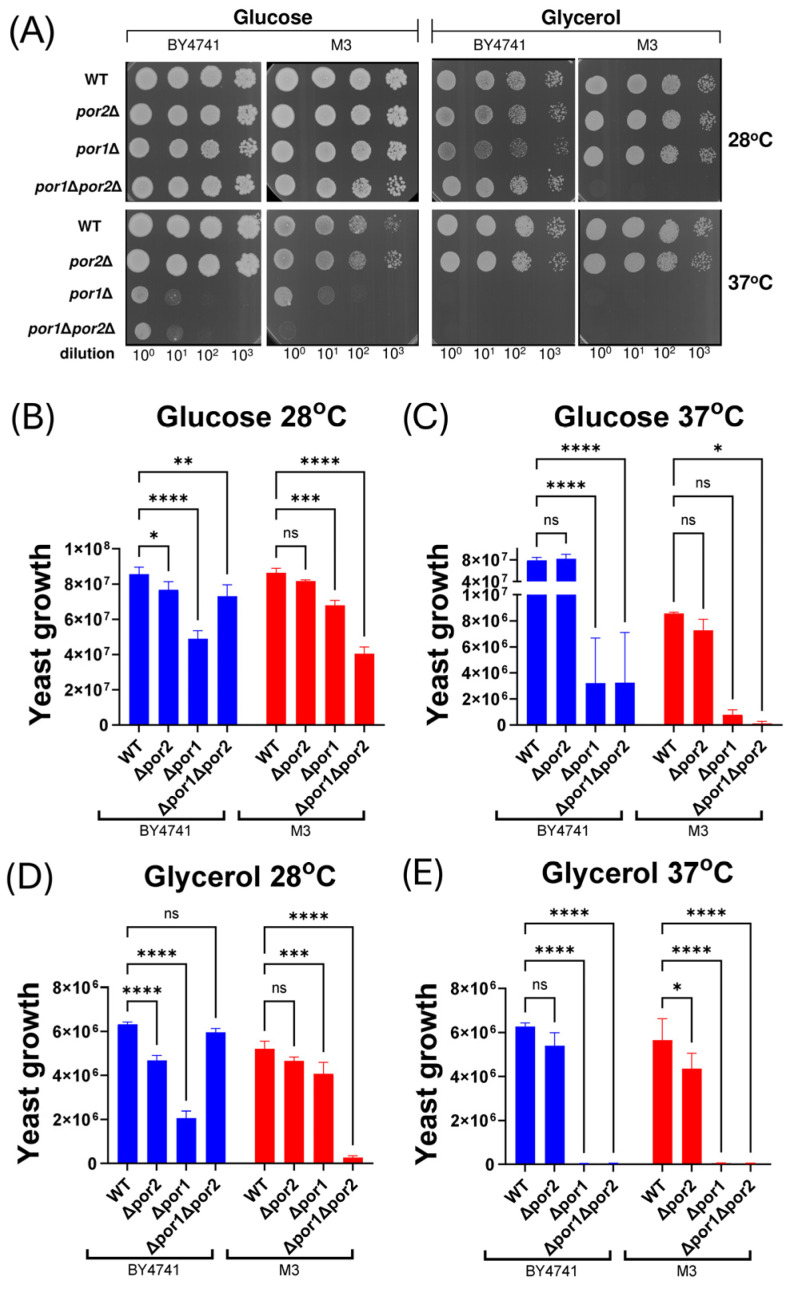
Results of the viability assay obtained for isogenic wild-type BY4741 and M3 strains, as well as for cells of *por1*Δ, *por2*Δ, and *por1*Δ*por2*Δ mutant strains. (**A**) Representative growth of BY4741 and M3 cells and cells of the relevant mutants on media supplemented with glucose (fermentable carbon source) or glycerol (nonfermentable carbon source) for 3 days at 28 °C (permissive temperature) and 37 °C (restrictive temperature). Serial tenfold dilutions of each of the yeast cell suspensions were spotted from left to right. Three replicates of the assay were performed. (**B**–**E**) Quantitative analysis of the growth of BY4741 and M3 cells and cells of the relevant mutants on media supplemented with glucose (**B**,**C**) or glycerol (**D**,**E**) at 28 °C (**B**,**D**) or 37 °C (**C**,**E**). The data are presented as the mean values ± standard deviations of three independent experiments. ns, not statistically significant; ****, *p* < 0.0001; ***, *p* < 0.001; **, *p* < 0.01; *, *p* < 0.05.

**Figure 3 ijms-25-13010-f003:**
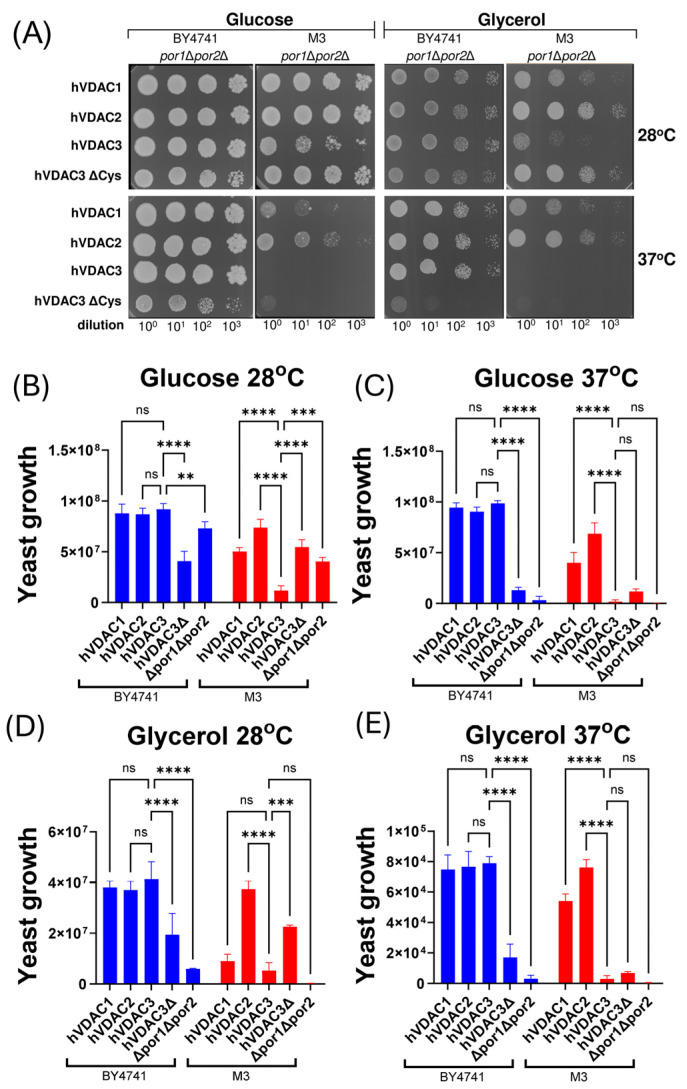
The effects of the heterologous expression of each human VDAC paralog or a cysteine-depleted variant of human VDAC3 on the results of the viability assay performed for M3-*por1*Δ*por2*Δ and BY4741-*por1*Δ*por2*Δ mutant cells. (**A**) Representative growth of the double mutant cells without and with expression of human VDAC paralogs (hVDAC1, hVDAC2, hVDAC3) or cysteine-depleted variant of hVDAC3 (hVDAC3ΔCys) on glucose (fermentable carbon source) or glycerol (non-fermentable carbon source) -containing media for 3 days at 28 °C (permissive temperature) and 37 °C (restrictive temperature). Serial tenfold dilutions of each of the yeast cells’ suspensions were spotted from the left to the right. Three repeats of the assay were performed. (**B**–**E**) Quantitative analysis of the growth of double mutant cells expressing human VDAC paralogs (hVDAC1, hVDAC2, hVDAC3) or a cysteine-depleted variant of hVDAC3 (hVDAC3ΔCys). These cells were grown on media supplemented with glucose (**B**,**C**) or glycerol (**D**,**E**) at 28 °C (**B**,**D**) or 37 °C (**C**,**E**). The quantitative analysis of the growth of double mutant cells not expressing human VDAC paralogs is based on data shown in Figure 2A. The data are presented as the mean values ± standard deviations of three independent experiments. ns, not statistically significant; ****, *p* < 0.0001; ***, *p* < 0.001; **, *p* < 0.01.

**Figure 4 ijms-25-13010-f004:**
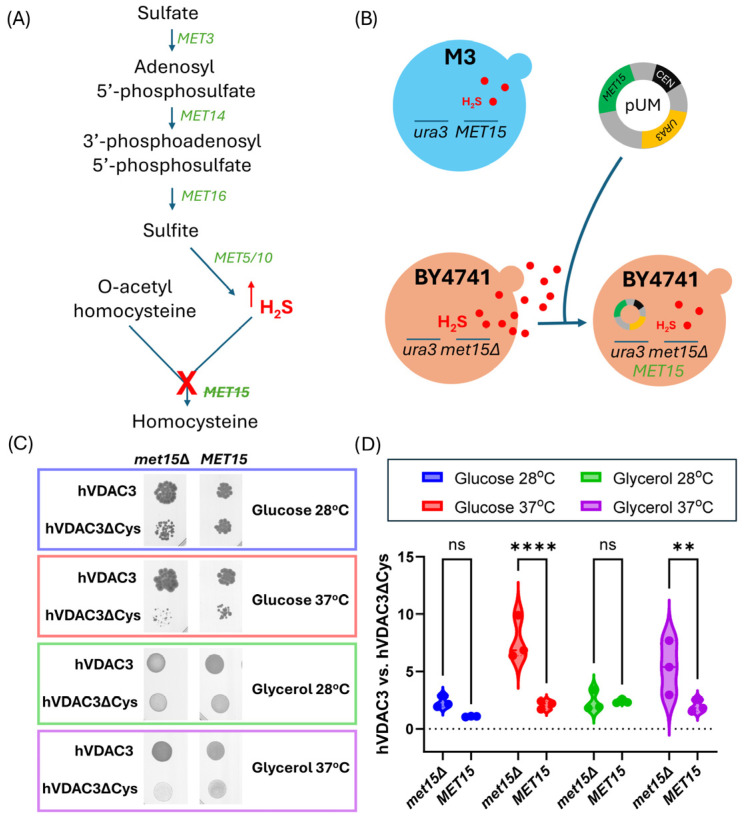
The effect of cysteine residues in hVDAC3 on the results of the viability assay in the presence or absence of the *MET15* auxotrophic marker. (**A**) Schematic illustration of the disruption of the sulfate assimilation pathway in *S. cerevisiae* cells due to *MET15* deletion in BY4741 cells. When the pathway is functional, sulfate (SO_4_^2−^) is enzymatically reduced to hydrogen sulfide (H_2_S), which is subsequently converted to homocysteine upon combination with O-acetyl homoserine. However, due to the *MET15* mutation, the pathway is interrupted, resulting in the accumulation of H_2_S. (**B**) Schematic illustration of the increase in H_2_S production in BY4741 cells caused by *MET15* deletion and the predicted decrease in H_2_S concentration following *MET15* gene complementation using the pUM vector. (**C**) Growth analysis of BY4741-*por1*Δ*por2*Δ mutant cells expressing hVDAC3 or hVDAC3ΔCys in the background of *MET15* deletion (*met15*Δ) or *MET15* complemented with the pUM plasmid (*MET15*), based on the viability assay results. *MET15* cells were grown on uracil-free medium supplemented with glucose or glycerol as a fermentable or nonfermentable carbon source, respectively, for 3 days at 28 °C (permissive temperature) or 37 °C (restrictive temperature). Three replicates of the assay were performed. The data for *met15*Δ cells are presented in Figure 3E. (**D**) Quantitative analysis of the growth of BY4741-*por1*Δ*por2*Δ mutant cells expressing hVDAC3 or hVDAC3ΔCys, focusing on the variation induced by the presence or absence of the *MET15* gene. The data are presented as violin plots of three independent experiments. ns, not statistically significant; ****, *p* < 0.0001; **, *p* < 0.01.

**Figure 5 ijms-25-13010-f005:**
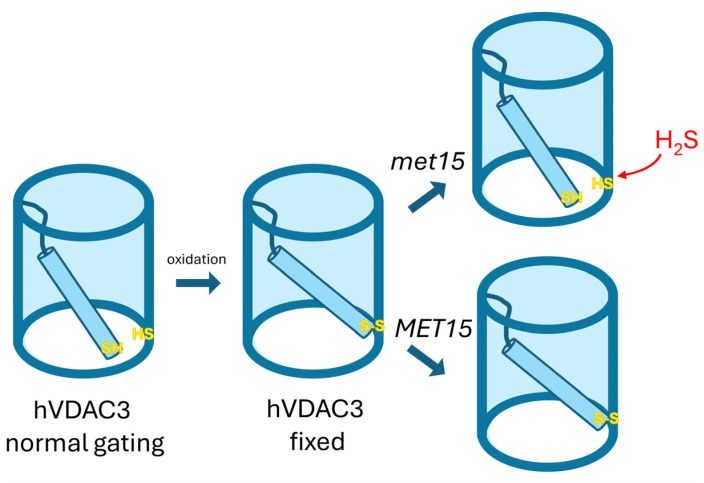
Influence of yeast genetic background on *MET15* and VDAC3 gating. In yeast strains with the recessive *met15* mutation, hydrogen sulfide is produced, which activates VDAC3 gating (VDAC3 normal gating). This gating is triggered by the suppression of disulfide-bond formation between the N-terminal region and the pore. Conversely, in strains with a functional *MET15* gene, hVDAC3 channels are likely stabilized in an open state under oxidizing conditions (hVDAC3 fixed). This hypothesis suggests that the genetic background of yeast strains, particularly regarding the *MET15* gene, can significantly influence the phenotypic outcomes related to hVDAC3 expression. Different genetic backgrounds may, therefore, result in distinct phenotypes due to varying mechanisms of VDAC3 gating activation and stabilization.

**Table 1 ijms-25-13010-t001:** List of yeast strains used in this study.

Strain	Genotype	Feature	Source
M3	*MATa lys2*; *his4*; *trp1*; *ade2*; *leu2*; *ura3*	WT	[17]
M3-*por1*Δ	*MATa lys2*; *his4*; *trp1*; *ade2*; *leu2*; *ura3*; *por1Δ::LEU2*	Lacking the *POR1* gene (*por1Δ*)	[17] (M22-2)
M3-*por2*Δ	*MATa lys2*; *his4*; *trp1*; *ade2*; *leu2*; *ura3*; *por2Δ::TRP1*	Lacking the *POR2* gene (*por2Δ*)	[17] (M3-2)
M3-*por1*Δ*por2*Δ	*MATa lys2*; *his4*; *trp1*; *ade2*; *leu2*; *ura3*; *por1Δ::LEU2*; *por2Δ::TRP1*	Double mutant lacking *POR1* and *POR2* genes (*por1Δ por2Δ*)	[17] (M22-2-1)
M3-*por1*Δ*por2*Δ-hVDAC1	*MATa lys2*; *his4*; *trp1*; *ade2*; *leu2*; *ura3*; *por1Δ::HVDAC1*; *por2Δ::TRP1*	Expresses human VDAC1 under the control of the *POR1* promoter (hVDAC1)	This work
M3-*por1*Δ*por2*Δ-hVDAC2	*MATa lys2*; *his4*; *trp1*; *ade2*; *leu2*; *ura3*; *por1Δ::HVDAC2*; *por2Δ::TRP1*	Expresses human VDAC2 under the control of the *POR1* promoter (hVDAC2)	This work
M3-*por1*Δ*por2*Δ-hVDAC3	*MATa lys2*; *his4*; *trp1*; *ade2*; *leu2*; *ura3*; *por1Δ::HVDAC3*; *por2Δ::TRP1*	Expresses human VDAC3 under the control of the *POR1* promoter (hVDAC3)	This work
M3-*por1*Δ*por2*Δ-hVDAC3-ΔCys	*MATa lys2*; *his4*; *trp1*; *ade2*; *leu2*; *ura3*; *por1Δ::HVDAC3ΔCys*; *por2Δ::TRP1*	Expresses human VDAC3 under the control of the *POR1* promoter. HVDAC3 mutations: C2A, C8A, C36A, C65A, C122A, C229A (hVDAC3ΔCys)	This work
BY4741	*MATa his3Δ1*; *leu2Δ0*; *met15Δ0*; *ura3Δ0*	WT	Euroscarf
BY4741-*por1*Δ	*MATa*; *his3Δ1*; *leu2Δ0*; *met15Δ0*; *ura3Δ0*; *por1Δ0*	Lacking the *POR1* gene (*por1Δ*)	This work
BY4741-*por2*Δ	*MATa*; *his3Δ1*; *leu2Δ0*; *met15Δ0*; *ura3Δ0*; *por2Δ::kanMX4*	Lacking the *POR2* gene (*por2Δ*)	Euroscarf
BY4741-*por1*Δ *por2*Δ	*MATa*; *his3Δ1*; *leu2Δ0*; *met15Δ0*; *ura3Δ0*; *por1Δ0*; *por2Δ::kanMX4*	Double mutant lacking *POR1* and *POR2* genes (*por1Δ por2Δ*)	This work
BY4741-*por1*Δ*por2*Δ-hVDAC1	*MATa*; *his3Δ1*; *leu2Δ0*; *met15Δ0*; *ura3Δ0 por1Δ::HVDAC1*; *por2Δ::kanMX4*	Expresses human VDAC1 under the control of the *POR1* promoter (hVDAC1)	This work
BY4741-*por1*Δ*por2*Δ-hVDAC2	*MATa*; *his3Δ1*; *leu2Δ0*; *met15Δ0*; *ura3Δ0*; *por1Δ::HVDAC2*; *por2Δ::kanMX4*	Expresses human VDAC2 under the control of the *POR1* promoter (hVDAC2)	This work
BY4741-*por1*Δ*por2*Δ-hVDAC3	*MATa*; *his3Δ1*; *leu2Δ0*; *met15Δ0*; *ura3Δ0*; *por1Δ::HVDAC3*; *por2Δ::kanMX4*	Expresses human VDAC3 under the control of the *POR1* promoter (hVDAC3)	This work
BY4741-*por1*Δ*por2*Δ-hVDAC3-ΔCys	*MATa*; *his3Δ1*; *leu2Δ0*; *met15Δ0*; *ura3Δ0*; *por1Δ::HVDAC3ΔCys*; *por2Δ::kanMX4*	Expresses human VDAC3 under the control of the *POR1* promoter. HVDAC3 mutations: C2A, C8A, C36A, C65A, C122A, C229A (hVDAC3ΔCys)	This work

**Table 2 ijms-25-13010-t002:** List of plasmids used in this study.

Plasmid	Size (bp)	Description
pML104-*POR1*	11,258	CRISPR/Cas9 vector designed for targeting the *POR1*
pBSK(+) Simple-Amp-*hVDAC1*	4383	Contains the repair DNA sequence for human *VDAC1*, used in CRISPR/Cas9-mediated gene editing
pBSK(+) Simple-Amp-*hVDAC2*	4416	Contains the repair DNA sequence for human *VDAC2,* used in CRISPR/Cas9-mediated gene editing
pBluescript II SK(+)-*hVDAC3*	4437	Contains the repair DNA sequence for human *VDAC3*, used in CRISPR/Cas9-mediated gene editing
pBluescript II SK(+)-*hVDAC3ΔCys*	4437	Contains the repair DNA sequence for cysteine-depleted variant of human *VDAC3*, used in CRISPR/Cas9-mediated gene editing
pUM	6448	Contains *URA3* and *MET17* auxotrophy selection markers

**Table 3 ijms-25-13010-t003:** List of oligonucleotides used in this study.

Oligonucleotides	Sequence (5′ -> 3′)	Description
pML104_por1_F	GTTTTAGAGCTAGAAATAGCAAGTTAAAATAAGGC	insertion of guide sgRNA sequence into pML104 vector
pML104_por1_R	GCAGTGAAAGATAAATGATCGATCATTTATCTTTCACTGC	
POR1A	TTCCAACAAGTTTAATGGTCAGAAT	amplification of repair DNA, sequencing, diagnostic
POR1B	CTCTAATTTGGTTTGCAAGTTGTTT	diagnostic
POR1C	AACTGCAAACTACCTAACTCCAATG	diagnostic
POR1D	AATGTTCGAAACCAATCTGAAAATA	amplification of repair DNA, sequencing, diagnostic
hVDAC1/2opt_B	CCGAATTCAGTACCGTCGTT	diagnostic
hVDAC3opt_B	GCCGGTGTTAGGAACAAAAA	diagnostic
POR1_KO_1	CCAACACGAAACAGCCAAGCGTACCCAAAGCAAAAATCAAA CCAACCTCTCAACAACGTATATATCTAATATATATATGTTC ACTATATACCATATATGTGCTCGTTCTT	oligonucleotides for hybridization, DNA repair for gene deletion
POR1_KO_2	AAGAACGAGCACATATATGGTATATAGTGAACATATATATA TTAGATATATACGTTGTTGAGAGGTTGGTTTGATTTTTGCT TTGGGTACGCTTGGCTGTTTCGTGTTGG	

## Data Availability

All data are contained within the article and Appendix A.

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
