# Peer review of "Developing a Novel and Optimized Yeast Model for Human VDAC Research"

_ijms, 2024, doi:10.3390/ijms252313010_

Round 1

Reviewer 1 Report

Comments and Suggestions for Authors

In this article, Baranek-Grabińska et al. use the yeast Saccharomyces cerevisiae to investigate the effect of the heterologous expression of VDAC paralogs (hVDAC1-hVDAC3) in a por1Δpor2Δ mutant strain. The results obtained by comparing this expression in two different yeast strains (M3 and BY4741) suggest that the influence of the yeast genotype is fundamental for this complementation assay. The study is undoubtedly interesting, but some major revisions are necessary to improve its scientific soundness.

MINOR REVISION

1) I think there is an issue with the layout of the text, as all figures appear before their citation. This makes reading the results and text very difficult.

2) Changing the order of the figures in panels 2 and 3, specifically showing the results of the spot assay analysis before the quantitative analysis, could improve the presentation of the results and their overall flow.

3) The authors describe three different wild-type strains (M3, BY4741, and BY4742) to support the reported results and highlight the importance of yeast genotypes in evaluating the capacity for complementation. Adding a sentence explaining why the BY4742 strain was not used in the experiments would clarify for the reader why it is described in such detail in the introduction.

4) Results Section 2.2, Line 175: It is not correct to refer to this assay as "cell viability," as the double mutant strain is viable. It would be more appropriate to describe this as an analysis to evaluate the capacity of the different hVDACs to complement the growth defect observed under fermentative or oxidative conditions.

MAJOR REVISION

FIG 2:

-            The phenotypes of the por1Δpor2Δ double mutants in BY4741 differ depending on the analysis performed: representative growth (Fig. 2E) versus quantitative analysis (Fig. 2B) - GLUCOSE 37°C conditions-. In one case, the por1Δ BY4741 strain shows lower growth levels compared to the por1Δpor2Δ double mutant, but this is not observed in the other. However, this observation is supported by the results obtained in Glycerol medium at 28°C. Why does the double mutant strain grow better than the single mutant strain?

-            The por1Δ M3 strain shows a "severe" growth defect in glucose at 37°C, which is not confirmed by the quantitative analyses. Could be discuss?

FIG 3:

Line 177 “expression of the paralogs in BY4741-por1Δpor2Δ mutant cells enhanced growth to a very similar extent”. I assume that this comment about “enhanced growth” is derived from comparing cell quantification in Fig 3A-B-C-D to Fig 2A-B-C-D. To make this analysis more significant and robust, I suggest including the corresponding double mutant strain in the comparison.

Author Response

MINOR REVISION

Comments 1:  I think there is an issue with the layout of the text, as all figures appear before their citation. This makes reading the results and text very difficult.

Response 1: We corrected layout accordingly.

Comments  2: Changing the order of the figures in panels 2 and 3, specifically showing the results of the spot assay analysis before the quantitative analysis, could improve the presentation of the results and their overall flow.

Response 2: We corrected the figures accordingly.

Comments  3: The authors describe three different wild-type strains (M3, BY4741, and BY4742) to support the reported results and highlight the importance of yeast genotypes in evaluating the capacity for complementation. Adding a sentence explaining why the BY4742 strain was not used in the experiments would clarify for the reader why it is described in such detail in the introduction.

Response 3: We agree with the Reviewer that the issue should be explained. Therefore, we changed the beginning of the last paragraph of the Introduction section as follows (start at line 113):

Given that BY4741-Δpor1 has never been used for heterologous expression of human VDAC paralogs and MET15 gene complementation in the strain cells can be easily performed by application of pUM vector we decided to perform heterologous expression of hVDAC1-hVDAC3 in BY4741 cells in the presence or absence of MET15. The analysis was to check putative relationship between yeast strain genotype background and hVDAC3 inability to complement the growth defect of M3-Δpor1 and BY4742-Δpor1 mutant cells under restrictive conditions [13,14].

Comments  4: Results Section 2.2, Line 175: It is not correct to refer to this assay as "cell viability," as the double mutant strain is viable. It would be more appropriate to describe this as an analysis to evaluate the capacity of the different hVDACs to complement the growth defect observed under fermentative or oxidative conditions.

Response 4: The sentence was corrected as follows:

To investigate whether an identical phenotype under restrictive conditions would result in similar effects of heterologous expression of human VDAC paralogs in both por1Δpor2Δ mutants, we evaluated the capacity of the different hVDAC paralogs to complement the growth defect observed under the applied conditions (Figure 3A).

MAJOR REVISION

Comments  5:  FIG 2:The phenotypes of the por1Δpor2Δ double mutants in BY4741 differ depending on the analysis performed: representative growth (Fig. 2E) versus quantitative analysis (Fig. 2B) - GLUCOSE 37°C conditions-. In one case, the por1Δ BY4741 strain shows lower growth levels compared to the por1Δpor2Δ double mutant, but this is not observed in the other. However, this observation is supported by the results obtained in Glycerol medium at 28°C.

Response 5: We thank the Reviewer for bringing this valid observation to our attention. Upon reevaluating of our data, we found that these differences are not statistically significant and fall within the range of experimental variation observed between repetitions. To minimize any potential confusion and to better indicate that there is no significant difference between these strains, we replaced the representative plate image with one that more accurately reflects the overall findings. Additionally, we performed a new quantitative analysis using a different dilution factor (10x) to ensure consistency and accuracy in our results.

 Comments  6:  Why does the double mutant strain grow better than the single mutant strain?

Response 6: This is a very interesting question. The better growth of BY4741-por1Δpor2Δ double mutant cells when compared with BY4741-por1Δ single mutant cells can be explained by genetic interaction that denotes modulation of the phenotype of one mutation by the presence of a second mutation (Collins et al., 2010). One explanation for this phenomenon is that the BY4741 strain can adapt to the complete absence of VDAC channels by upregulating alternative transport systems or reducing metabolic demands, thus compensating for the loss. In contrast, the M3 strain may lack such adaptive responses, making the loss of both POR1 and POR2 more detrimental to its growth.

Taking into account what is written above we introduced the following changes in the Discussion section (line 257):

The observed cell growth differences between the double mutant BY4741-por1Δpor2Δ and the single mutant BY4741-por1Δ reveal interesting genetic interactions that shed light on the functional relationship between POR1 and POR2. According to Collins et al. (2010), genetic interactions can be positive (suppressive) or negative (synthetic sick/lethal), depending on how mutations affect cellular processes. The positive genetic interaction observed for cells of BY4741 strain suggests that the absence of POR2 attenuates the deleterious effects caused by the deletion of POR1. This could be due to a compensatory mechanism whereby the complete loss of VDAC channels causes the cell to activate alternative metabolic pathways or stress responses to mitigate the impact on growth. In other words, the cell may reprogram its metabolism to compensate for the loss of VDACs. Conversely, in cells of M3 strain, the negative genetic interaction occurs because the relevant double mutation is more deleterious than the single POR1 mutation that indicates that POR1 and POR2 have independent but complementary roles that are critical for the cell growth of the strain. The lack of effective compensatory mechanisms in cells of M3 strain could exacerbate the negative effects of the loss of both VDAC channels.

Comments  7:  The por1Δ M3 strain shows a "severe" growth defect in glucose at 37°C, which is not confirmed by the quantitative analyses. Could be discuss?

Response 7: The Reviewer is right that the por1Δ M3 strain exhibits a severe growth defect on glucose at 37°C in the plate images, which was not initially reflected in our quantitative analyses. Upon reviewing our data, we realized that the dilution factor used in the quantitative analysis was not optimal for capturing the significant growth differences observed on the plates. Similar to the adjustments made in response to the previous comment, we  performed a new quantitative analysis using a different dilution factor (10x) that more accurately represents the actual growth levels of the yeast strains. This revised analysis aligns better with the qualitative results observed on the plates.

We appreciate your attention to this detail, which has allowed us to improve the accuracy and consistency of our reported results. We updated Figure 2 and corrected a relevant part of the Discussion section; line 257).

Comments  8:  FIG 3: Line 177 “expression of the paralogs in BY4741-por1Δpor2Δ mutant cells enhanced growth to a very similar extent”. I assume that this comment about “enhanced growth” is derived from comparing cell quantification in Fig 3A-B-C-D to Fig 2A-B-C-D. To make this analysis more significant and robust, I suggest including the corresponding double mutant strain in the comparison.

Response 8: We added the relevant double mutants to Figure 3.

Reviewer 2 Report

Comments and Suggestions for Authors

I do appreciate the work achieved by this group.

I do find the work meritorious of publication.

I also find the Abstract and the Conclusions of the work, WEAK.

Specifically, after all the work done the Abstract only reads, as a conclusion:

"Here, we applied por1Δpor2Δ mutants derived from 14

two S. cerevisiae strains commonly used for heterologous expression of hVDAC paralogs and 15

addressed the possible relationship between differential ability of hVDAC3 to reverse the growth 16

phenotype of the por1Δpor2Δ mutants and the presence of hydrogen sulfide (H2S)." THIS SEEMS CUT SHORT OR MID SENTENCE. WHAT DID YOU FIND? IS THIS A GOOD MODEL? SHOULD YOU USE A DIFFERENT APPROACH TO STAY VDAC3?

"The results 17

indicate the need to consider genotype background in application of yeast complementation assays 18

in research on the complexity of human VDAC paralogs, particularly human VDAC3. 19" THIS AGAIN MISSES THE OPPORTUNITY TO EXPAND ON THE FRUITS OF THE WORK.

In the Discussion, for example, the authors write: "Thus, H2S is emerging as a molecule im- 286

portant for modulation of hVDAC3 involvement in cellular response to oxidative stress 287

that is based on the protein cysteine residues but contributing mechanism(s) requires fur- 288

ther studies. Our hypothesis addressing the mechanism is shown in Figure 5. 289"

This is a conclusion of the work presented.

I suggest the authors review their conclusions to drive readers to understand and appreciate the opportunities that this work offers for future research.

Author Response

Comments  1:  I also find the Abstract and the Conclusions of the work, WEAK.

Specifically, after all the work done the Abstract only reads, as a conclusion: "Here, we applied por1Δpor2Δ mutants derived from two S. cerevisiae strains commonly used for heterologous expression of hVDAC paralogs and addressed the possible relationship between differential ability of hVDAC3 to reverse the growth phenotype of the por1Δpor2Δ mutants and the presence of hydrogen sulfide (H2S)."

THIS SEEMS CUT SHORT OR MID SENTENCE. WHAT DID YOU FIND? IS THIS A GOOD MODEL? SHOULD YOU USE A DIFFERENT APPROACH TO STAY VDAC3?

"The results indicate the need to consider genotype background in application of yeast complementation assays in research on the complexity of human VDAC paralogs, particularly human VDAC3."

THIS AGAIN MISSES THE OPPORTUNITY TO EXPAND ON THE FRUITS OF THE WORK.

Response 1: We corrected the Abstract accordingly by introducing the following changes starting from the indicated line 14:

Here, we applied por1Δpor2Δ double mutants and relevant por1Δ and por2Δ single mutants, derived from two S. cerevisiae strains (M3 and BY4741) differing distinctly in auxotrophic markers but commonly used for heterologous expression of hVDAC paralogs, to study the effect of the presence of yVDAC2 and cell genotype including MET15, the latter resulting in low level of hydrogen sulfide (H2S), on complementation potential of heterologous expression of hVDAC paralogs. The results indicated that yVDAC2 might contribute to the complementation potential. Moreover, the possibility to reverse the growth phenotype by heterologous expression of hVDAC paralogs in the presence of the applied yeast cell genotype backgrounds was particularly diverse for hVDAC3 and depended on the presence of the protein cysteine residues and expression of MET15. Thus, the difference in the set of auxotrophic markers in yeast cells, including MET15 contributing to H2S level, may create a different background for the modification of cysteine residues in hVDAC3 and thus explain the different effects of the presence and deletion of cysteine residues in hVDAC3 in M3-Δpor1Δpor2 and BY4741-Δpor1Δpor2 cells. The different phenotypes displayed by BY4741-Δpor1Δpor2 and M3-Δpor1Δpor2 cells following heterologous expression of particular hVDAC paralog make them valuable models for the study of human VDAC proteins, especially hVDAC3 as a representative of VDAC protein sensitive to reduction-oxidation state.

Comments  2:  In the Discussion, for example, the authors write: "Thus, H2S is emerging as a molecule important for modulation of hVDAC3 involvement in cellular response to oxidative stress that is based on the protein cysteine residues but contributing mechanism(s) requires further studies. Our hypothesis addressing the mechanism is shown in Figure 5."

THIS IS A CONCLUSION OF THE WORK PRESENTED.

I SUGGEST THE AUTHORS REVIEW THEIR CONCLUSIONS TO DRIVE READERS TO UNDERSTAND AND APPRECIATE THE OPPORTUNITIES THAT THIS WORK OFFERS FOR FUTURE RESEARCH.

Response 2: We agree with the Reviewer suggestion to strengthen our conclusions to help readers understand and appreciate the opportunities our work offers for future research. Thus, we removed two sentences at the end of the Discussion section and significantly revised Section 5 (lines 392-412) of the manuscript, where we newly formulated our conclusions based on the Reviewer insights. Consequently, the part was changed as follows:

In conclusion, our study demonstrates that the cysteine residues of hVDAC3 are crucial for its ability to complement growth defects in yeast strains lacking endogenous VDAC proteins, with this effect being modulated by the yeast's genetic background—particularly the presence or absence of the MET15 gene. The differential growth responses observed between M3-por1Δpor2Δ and BY4741-por1Δpor2Δ cells upon expression of hVDAC3 and its cysteine-depleted variant (hVDAC3ΔCys) suggests that hydrogen sulfide (H₂S) production plays a significant role in modulating hVDAC3 function. Specifically, in met15Δ strains like BY4741, increased H₂S production may enhance the activity of hVDAC3 by preventing disulfide bond formation involving cysteine residues, thereby influencing the hVDAC3 gating.

Our findings highlight H₂S as an emerging molecule important for the modulation of hVDAC3 involvement in the cellular response to oxidative stress, acting through modifications of cysteine residues. This opens new avenues for future research to explore the molecular mechanisms underlying this modulation. Understanding these mechanisms could provide valuable insights into the regulation of VDAC3 in eukaryotic cells and its potential role in conditions associated with oxidative stress. Our proposed hypothesis, illustrated in Figure 5, suggests that the genetic background of yeast strains, particularly regarding the MET15 gene, can significantly influence the phenotypic outcomes related to hVDAC3 expression by affecting VDAC3 gating mechanisms. Further studies are needed to validate this hypothesis and to investigate the potential applications of modulating VDAC3 activity in therapeutic contexts.

Round 2

Reviewer 1 Report

Comments and Suggestions for Authors

Honestly, I would have appreciated if the control shown in Figure 3 had also been included in the spot assay analysis. That said, I recognize the robustness of the data and appreciate the additional insights you have provided, so this works for me